# Marine Cloud Brightening of Cumulus Clouds: From the Sprayer to the Cloud

Johannes Kainz<sup>1</sup>, Daniel P. Harrison<sup>2</sup>, and Fabian Hoffmann<sup>3</sup>

Correspondence: Fabian Hoffmann (f.hoffmann@fu-berlin.de)

**Abstract.** Marine Cloud Brightening (MCB) is a suggested solar radiation management approach to mitigate global warming by increasing the reflectance of clouds through the emission of additional aerosols. While stratocumulus are considered the preferred target for MCB, the present study investigates trade-wind cumulus clouds, which may be the dominant cloud type for certain regional MCB deployments. In this study, high-resolution large-eddy simulations with detailed Lagrangian cloud microphysics are used to assess the role of different aerosol sprayer heights on the efficacy of MCB. The study indicates that surface sprayers are the optimal placement, as they facilitate the most efficient dispersion of aerosol within the boundary layer, which increases the fraction of clouds affected by the sprayed aerosols, as well as the transport of the sprayed aerosols into the developing clouds, which increases the number of cloud droplets developing from the sprayed aerosols.

### 1 Introduction

The proposed solar radiation management (SRM) approach Marine Cloud Brightening (MCB) aims to enhance cloud reflectance by modifying cloud microphysical properties with the aim to increase Earth's albedo and hence counter global warming (Latham, 2002; Latham et al., 2012; Feingold et al., 2024). Artificial cloud condensation nuclei (CCN) are introduced into clouds, redistributing liquid water onto a greater number of smaller droplets, thereby increasing the ability of clouds to reflect short-wave radiation back into space (Twomey, 1974, 1977). This approach has been predominantly studied for marine stratocumulus clouds, which are particularly susceptible to MCB due to their relatively small vertical extent and high cloud fraction (Wood, 2021; Zhang and Feingold, 2023).

The regional application of MCB is suggested to protect high-value local ecosystems from the adverse effects of global warming, such as the intensification and increased frequency of marine heatwaves affecting the Great Barrier Reef (GBR) in Australia. A proposed strategy to mitigate coral bleaching, a phenomenon triggered by heat stress, is to cool the reef by increasing cloud reflectance through a deployment of MCB, thereby reducing the solar radiation absorbed by the ocean (Harrison, 2024). In contrast to the original concept of MCB targeted at stratocumulus clouds, the GBR is dominated by scattered shallow cumulus clouds, where many findings for stratocumulus clouds may not apply.

To evaluate the feasibility and effectiveness of MCB under these conditions, we conduct high-resolution large-eddy simulations (LESs) with a detailed Lagrangian cloud microphysics model (LCM) (Hoffmann et al., 2015). The primary focus is on

<sup>&</sup>lt;sup>1</sup>Meteorological Institute, Ludwig-Maximilians-Universität München, München, Germany

<sup>&</sup>lt;sup>2</sup>National Marine Science Center, Southern Cross University, Coffs Harbour, Australia

<sup>&</sup>lt;sup>3</sup>Institute of Meteorology, Freie Universität Berlin, Berlin, Germany

the sprayed CCN transport, which are commonly suggested to be injected into the lower, cloud-free marine boundary layer from spraying apparatus mounted on boats. Another deployment strategy suggests the use of drones, to spray directly within the cloud layer (Claudel et al., 2024). Thus, we test three different sprayer heights: a sprayer positioned near the surface, a sprayer near the cloud base, and a sprayer positioned above the cloud layer. By leveraging the unique capability of the LCM to track individual aerosol particles, we quantify the transport of sprayed CCN to the clouds and examine how the sprayer height influences this process.

This study is organized as follows. First, the simulation setup including necessary model improvements and environmental conditions of the simulated case are discussed. Secondly, the results are analyzed and discussed with special emphasis on the cloud microphysical response to spraying in various heights, the plume dispersion, and the activation rate based on the sprayer position. Finally, the study is summarized and discussed.

#### 5 2 Simulation Setup

For this MCB study, a coupled modeling approach is applied. The dynamical core is the LES model System for Atmospheric Modeling (SAM) (Khairoutdinov and Randall, 2003), two-way coupled with LCM cloud microphysics (Hoffmann et al., 2017; Hoffmann and Feingold, 2019).

A non-precipitating trade-wind cumulus case is simulated, based on the seminal BOMEX intercomparison case (Siebesma et al., 2003). The environmental setup of the BOMEX case resembles trade-wind cumulus conditions frequently found over the GBR (Eckert et al., 2023). Figure 1 shows four soundings from Willis Island, located upwind of the central GBR, from December 10-12, 2022. Especially the thermodynamic quantities agree well with the domain-averaged LESs results.

The modeling domain measures  $20,480\,\mathrm{m} \times 5,120\,\mathrm{m} \times 20,480\,\mathrm{m}$ , with a grid box size of  $40\,\mathrm{m} \times 40\,\mathrm{m} \times 20\,\mathrm{m}$  in the two horizontal and the vertical direction, respectively. The aerosol sprayer is positioned in the center of the short edge of the xy-plane, with the long edge aligned parallel to the mean wind direction. This domain configuration enables plume analysis up to an hour downwind.

The LCM represents cloud microphysics by simulating Lagrangian computational particles, each representing a multitude of real aerosol particles or cloud droplets with identical properties, e.g., water mass and aerosol mass. Condensation and evaporation of wetted aerosols and cloud droplets is represented by solving the diffusion equation for each LCM particle individually (Hoffmann et al., 2015). Moreover, the particles experience sedimentation (Beard, 1976). To maintain the identity of sprayed CCN, collectional growth was disabled, and is also not expected due to the relatively high droplet concentrations. It is important to note that the LCM treats all hydrometeors in a uniform manner. The categorization into aerosol particles and cloud droplets is undertaken solely during the analysis process. Additionally, the LCM also provides the ability to track sprayed aerosols and cloud droplets, which enables a straight-forward way to infer some effects of the spraying on the simulated clouds. For more details on the implementation of the LCM, see Hoffmann et al. (2015).

In each LES gridbox, 50 LCM particles are placed, representing the lognormally distributed background aerosol of a concentration of 200 cm<sup>-3</sup>, a geometric mean aerosol radius of 50 nm, and a geometric standard deviation of 1.25, representing

Figure 1. The left panel displays the domain-averaged profiles of liquid water potential temperature  $\theta_l$  (K) (blue) and the total water mixing ratio  $q_t$  (red). The right panel shows the mean horizontal wind. Thin lines depict data from individual soundings, thick straight lines show the average of the soundings, and the dashed lines show the LES domain-averaged data. Shaded grey areas mark the cloud layer.

typical marine conditions found over the GBR (Horchler et al., 2025; Braga et al., 2025). Sprayed LCM particles are placed in a predefined gridbox with a rate of 5 LCM particles per timestep, similar to Prabhakaran et al. (2023), but with a higher number of LCM particles added per gridbox and timestep. The sprayed aerosols are also lognormally distributed with a geometric mean radius of sprayed aerosols of 40 nm, a geometric standard deviation of 2.0, and a seeding rate of 10<sup>14</sup> aerosol particles per second. The aerosol initial wet radius is chosen corresponding to 3.5% NaCl in sea water. These parameters are specifically chosen to represent the current technological feasibility of potential MCB sprayers (Hernandez-Jaramillo et al., 2023). However, it should be noted that this current state of technology is not yet matching the theoretically proposed ideal spraying parameters, especially a spraying rate of 10<sup>15</sup> particles per second (cf. Wood, 2021).

Three distinct simulation setups are performed, each corresponding to a different aerosol sprayer altitude: one at the sea surface (30 m), another slightly above the cloud base (810 m), and a third near the cloud top (1810 m). These positions are representative for a surface, a cloud-base, and a cloud-top sprayer. A six-hour spin-up period precedes the injection of aerosols, followed by one hour of simulation time that is used for analysis. We avoid longer analysis because the plume would interfere with itself due to the cyclic boundary conditions. Each setup comprises ten ensemble members, each with slightly perturbed initial thermodynamic conditions to mitigate the limited time for analysis of the sprayed clouds and the natural variability in the cumulus cloud field.

#### 3 Results and Discussion

The qualitative behavior of the three different sprayer heights is displayed in Fig. 2, showing a snapshot from the first ensemble run for each scenario. A top view is compared to a profile cut, with colors indicating the average sprayed aerosol concentration, and continuous black lines encircling the clouds. While the top view averages the aerosol concentration over the entire vertical column, the horizontal averaging is restricted to the region between the dashed black lines indicated in the left column, which follow the mean wind and hence the center of the plume for each sprayer height.

In Figs. 2a, c, and f one sees that the height of the sprayer and thus the sprayed aerosol do not substantially affect the overall morphology of the cumulus cloud field. This is to be expected because adjustments of the cloud field tend to require much more time to be effective (e.g., Glassmeier et al., 2021; Chen et al., 2024). Nonetheless, the development of the plume changes substantially with the sprayer height. First, the increasing static stability depletes turbulence with height (Figs. 1 and 4), which reduces the horizontal expansion of the aerosol plume sprayed at greater heights. In addition, the horizontal wind slows down with height (Fig. 1), which reduces the distance the plume travels in a fixed time period. Due to the resultant lower dilution, higher sprayed aerosol concentrations are simulated for higher altitude sprayers.

Figures 2b, d, and f show the importance of clouds for the vertical transport of the sprayed aerosol. Whenever the plume interacts with a cloud, the sprayed aerosol experiences strong vertical transport and is rapidly mixed throughout the cloud volume. Particularly, aerosol sprayed in the sub-cloud layer benefit from updrafts that transport the aerosols to the base of developing cumulus clouds, as can be seen by the overlapping cloud contours and plume regions of the two large clouds at 7,000 and 14,500 m distance downwind (Fig. 2b).

**Figure 2.** Horizontal (left column) and vertical (right column) cross-sections of the average sprayed aerosol concentration (green-blue) and cloud location (black contours) for a surface sprayer (a, b), cloud-base sprayer (c, d), and cloud-top sprayer (e, f). The concentrations are averaged over the entire vertical axis in case of the horizontal cross-sections, while the vertical cross-sections use the volume between the dashed lines shown in the left column. The mean concentration of the of the horizontal-cross section is indicated by a single tick within the color bar.

### 3.1 Cloud Optical Response

We assess the cloud radiative effect (rCRE) (Betts, 2007) to determine the efficiency of MCB (Hoffmann and Feingold, 2021). The rCRE can be approximated as the product of the cloud albedo  $A_c$  and cloud fraction  $f_c$ ,

$$rCRE = f_c A_c. (1)$$

Assuming overhead sun, we determine

$$A_{\rm c} = \frac{\tau_{\rm c}(1-g)}{\tau_{\rm c}(1-g) + 2},\tag{2}$$

Figure 3. Probability density functions (PDFs) of cloud Albedo  $A_c$ , liquid water path (LWP), and droplet number mixing ratio  $(N_c)$  for all three sprayer heights (lines) are displayed. The right-most column shows the corresponding cloud fractions  $(f_c)$ . Gray, red, and blue lines indicate values for the total domain and regions affected by the sprayed aerosol or not, respectively. The subscript \* refers to the clouds containing sprayed aerosol, and subscript 0 refers to the remaining, unaffected clouds. Vertical lines indicate the respective averages. All ensemble simulations are used to determine these data.

120

(e.g., Bohren, 1987), with g the asymmetry parameter and

$$\tau_{\rm c} = \int \left[ \int Q_{\rm ext}(r,\lambda) \pi n(r,z) dr \right] dz \sim \text{LWP}_c^{5/6} N_c^{1/3}, \tag{3}$$

where n(r,z) is the particle size distribution,  $Q_{\rm ext}$  the extinction coefficient,  $\lambda$  the wavelength, r the particle radius, LWP $_c$  the vertically integrated liquid water content (liquid water path), and  $N_c$  the droplet number concentration. We use  $\lambda = 500\,\mathrm{nm}$ , representative of the bulk of solar shortwave radiation. The inclusion of (1-g) accounts for the increased backward scattering for small particles (Kokhanovsky, 2004).

We show  $A_c$  along LWP<sub>c</sub> and  $N_c$  in the first three columns of Fig. 3, followed by  $f_c$  in the right-most column to give a comprehensive overview of the parameters determining the rCRE for different sprayer heights (rows).

 $A_c$  for all clouds (gray lines, no superscript), sprayed clouds (red lines, asterisk superscript), and non-sprayed clouds (blue, 0 superscript) is displayed in Figs. 3a, e, and i. A cloud is considered sprayed when there is at least one sprayed LCM particle in a vertical column. As one expects,  $A_c$  of the unsprayed clouds is largely unaffected by the sprayer height, and the sprayed clouds seem to exhibit larger average  $A_c$  (indicated by vertical lines). However,  $A_c$  for sprayed clouds increases with sprayer height. What causes this behavior?

Figures 3b, f, and j present the LWP<sub>c</sub> for different sprayer heights. We see that the LWP<sub>c</sub> peaks around 50 g m<sup>-2</sup> for all clouds, with a long tail towards high values, depecting rare deep clouds. For the surface sprayer, LWP<sub>c</sub> in the sprayed clouds is nearly unchanged compared to all clouds, indicating that clouds of all sizes are affected by the sprayer. In contrast, higher sprayers primarily impact deeper, high-LWP<sub>c</sub> clouds, while the LWP<sub>c</sub> of all clouds is similar to the lower sprayer heights. This indicates that the increase in the average LWP<sub>c</sub> in sprayed clouds is not due to the spraying creating deeper clouds but due to the sprayed aerosol interacting preferentially with deeper clouds.

Figures 3c, g, and k show  $N_c$ . The intended increase in  $N_c$  is most pronounced for the surface sprayer. As we will discuss in Sec. 3.3., this is due to a higher probability of sprayed aerosols to activate to cloud droplets if they enter the clouds through the cloud base (surface sprayer) than by lateral entrainment (higher sprayers) (Hoffmann et al., 2015; Oh et al., 2023). In fact,  $N_c$  barely changes for higher sprayers compared to the non-sprayed clouds, and even slightly decreases for the highest sprayer, which might be due to the stronger dilution of deeper clouds interacting with the sprayed aerosol.

The behavior of LWP<sub>c</sub> and  $N_c$  with sprayer height indicates that the increase in  $A_c$  for the surface sprayer is mainly due to an increase in  $N_c$ , the main mechanism of MCB (Twomey, 1974). The increase in  $A_c$  for higher sprayers, however, is due to the sprayed aerosol primarily interacting with deeper and hence higher LWP<sub>c</sub> clouds, and is therefore not a result of MCB.

Figures 3d, h, and 1 show that the area fraction of clouds affected by the plume,  $f_c^*$ , decreases with sprayer altitude. As already indicated for Fig. 2 and further discussed in Sec. 3.2 below, the decrease in  $f_c^*$  is due to weaker turbulence limiting plume dispersion for higher sprayers.

How does this affect the rCRE? Assuming that  $f_c$  remains unchanged due to spraying (cf. 3d, g, and l), the resultant change in the rCRE due to spraying is

$$d(rCRE) = f_c^* (A_c^* - A_c^0). \tag{4}$$

Figure 4. The domain-averaged vertical profile of the TKE from the LES simulations is shown. Shaded grey areas mark the cloud layer.

Based on this metric, the surface sprayer shows the largest brightening [d(rCRE)=0.0021], while it decreases with height [d(rCRE)=0.0019 and d(rCRE)=0.0006]. Compared to the base rCRE=0.2766, the change is small. We will discuss how to increase d(rCRE) in Sec. 4.

## 3.2 Plume Dispersion

A comprehensive understanding of plume propagation is important for optimizing the deployment of MCB, as has been discussed recently by Dhandapani et al. (2025), who simulated a single surface sprayer's plume dispersion for MCB deployment, using LESs with a bulk microphysics scheme. In the trade-wind cumulus region, a well-mixed, decoupled sub-cloud layer exists between the ocean surface and cloud base (here at 500 m), with a conditionally stable cloud layer above, followed by an

unconditionally stable free troposphere (here 1,500 m), as indicated by the temperature profile in Fig. 1. The increasing stability with height depletes turbulence (Siebesma et al., 2003; Stull, 1988). The vertical profile of turbulence kinetic energy (TKE) in this shallow cumulus environment reflects the combined influence of surface forcing, buoyancy, and stratification (Fig. 4). Near the surface, TKE production is high due to strong mechanical generation from shear and convective turbulence driven by surface heating, resulting in vigorous mixing in the sub-cloud layer. With height, mechanical production gradually declines as the influence of the surface weakens. Within the cloud layer, TKE stays constant due to buoyancy production associated with latent heat release during condensation, enhancing turbulent mixing within the clouds. Above the cloud tops, TKE diminishes where turbulence is dampened by the stable stratification. This vertical structure underlies the observed plume dispersion patterns.

The development of the mean horizontal plume width and the corresponding minima and maxima are shown in Figs. 5a, b, and c. The sub-cloud layer experiences the strongest turbulent dispersion, evident by the rapid increase in mean plume width downwind of the surface sprayer (Fig. 5a), while the dispersion is clearly slower at higher levels (Figs. 5b and c), reflecting the aformentioned role of turbulence in widening the aerosol plume (Fig. 4).

Figures 5d, e, and f show the vertical plume dispersion. While the plume emitted by the surface sprayer remains for some time in the sub-cloud layer, it reaches the cloud layer (on average) about 5,000 m downwind of the sprayer. In the cloud layer, the plume quickly disperses in the vertical by clouds lifting the aerosols up to 1,800 m. This vertical transport within the cloud layer is similar for higher sprayers. However, the transport of aerosol from the higher sprayers to the sub-cloud layer is slow, indicating that a large fraction of the aerosol sprayed from higher sprayers remains in the cloud layer.

The dispersion patterns for the three sprayer heights suggest that surface sprayers are the most effective for seeding a larger region, while cloud-top sprayers are the least efficient. This is attributed to more extensive horizontal dispersion at lower altitudes, allowing aerosols to be entrained into multiple updrafts over a broader region, which is in good agreement with the result from Figs. 3d, h, and l, which show that the highest  $f_c^*$  is found for the surface sprayer.

### 3.3 Activation Rate

In Fig. 6, the fraction of sprayed aerosols that are activated to cloud droplets is displayed for the surface sprayer (green), the cloud-base sprayer (orange), and the cloud-top sprayer (blue). Only regions with sprayed aerosols inside the cloud (liquid water content  $> 0.01 \,\mathrm{g\,m^{-3}}$ ) are considered.

The surface sprayer leads to the highest activation fraction, while the higher sprayers in the cloud layer tend to show an overall reduced activation fraction. As indicated by Fig. 2 and Figs. 5 d, e, and f, aerosols emitted from the surface sprayer enter the cloud through the cloud base, where the largest supersaturations occur, enabling a large fraction of aerosols to activate to cloud droplets. For higher sprayers, the way to activation is primarily through lateral entrainment. Because supersaturations are significantly lower for this pathway (Hoffmann et al., 2015; Oh et al., 2023), activation is not as effective as for the surface sprayer.

Why does the activation fraction increase with distance to the sprayer? While the fraction of aerosols transported from higher sprayers into the sub-cloud layer increases with distance to the sprayer (Figs. 5 e and f), more aerosols can activate at cloud base and the activation fraction of higher sprayers increases, just as for the surface sprayer. At the same time, the aerosol plume

Figure 5. For three sprayer heights, the plume dispersion in the horizontal plane (upper panels) and vertical plane (lower panels) are displayed. For the horizontal extension the mean plume width (black) and its maximum (red) and minimum (blue) extend are shown as a function of the distance to the sprayer. For the vertical extent the mean upper (red) and mean lower (blue) height relative to the sea surface, as well as the maximum upper (pale red) and minimum lower (pale blue) height are shown. The ensemble data is averaged over 35 min, with the tip of the plume (first 15% of the streamwise development) excluded to prevent a bias due to its lower horizontal development.

**Figure 6.** The averaged sprayed aerosol activation fraction for the surface sprayer (green), cloud-base sprayer (orange) and cloud-top sprayer (blue) are displayed as a function of the distance from the sprayer.

dilutes (Fig. 5), which decreases the competition for water vapor during activation, which also allows more aerosol particles to activate (Hernandez-Jaramillo et al., 2023).

#### 4 Conclusions

Marine cloud brightening (MCB) is a proposed strategy to mitigate the effects of global warming by increasing the reflectance of clouds. This strategy is based on increasing the number concentration of cloud droplets by adding additional aerosols (Latham et al., 2012; Feingold et al., 2024). While prior studies predominantly centered on marine stratocumulus clouds due to their heightened susceptibility to MCB (e.g., Dhandapani et al., 2025; Hoffmann and Feingold, 2021), this study investigates MCB for trade-wind cumulus clouds to evaluate its potential for cooling small, high-value sites such as the Great Barrier Reef (GBR) in Australia, as recently proposed by Harrison (2024).

To address open points regarding this proposed strategy, we performed high-resolution large-eddy simulations with detailed Lagrangian cloud microphysics. The impact of a single aerosol sprayer, placed at three heights, on a shallow cumulus field has been analyzed, showing that MCB generally is applicable under such conditions. The choice of sprayer height has a substantial impact on the increase of the cloud albedo of the region affected by the sprayed aerosol, with the surface sprayer resulting in the largest increase in the relative cloud radiative effect (rCRE). There are two main reasons for this:

- Surface aerosol spraying benefits from the strongest plume dispersion, caused by more vigorous turbulence in the subcloud layer. In contrast, higher sprayers experience reduced plume dispersion due to weaker turbulence, leading to a smaller sprayed cloud fraction.

- Aerosols introduced at the surface exhibit the highest activation fraction, that is the concentration of activated droplets produced from sprayed aerosols, as they are transported into clouds through their base by updrafts connecting subcloud and cloud layer. The strong supersaturations experienced at cloud base result in a great number of newly formed cloud droplets, which is the primary mechanism driving cloud brightening (Latham et al., 2012). In contrast, aerosols introduced at higher altitudes primarily enter clouds by lateral entrainment, where lower supersaturations limit their ability to activate, reducing the effectiveness of the seeding process (Hoffmann et al., 2015; Oh et al., 2023).

Thus, the increase in the rCRE is strongest for surface sprayers, which seed a larger fraction of the cloud field and enhance 195 cloud reflectivity through more efficient aerosol activation. Mid- and high-altitude sprayers show reduced efficiency due to limited plume dispersion and lower seeding effectiveness. These findings indicate that for cumulus environments, surface sprayers offer the most viable MCB deployment strategy.

Our results emphasize the importance of optimizing aerosol injection strategies for effective MCB implementation in cumulus regions. As our study focused on the first 50 minutes after spraying, future studies should further investigate the long-term interactions between sprayed clouds and broader atmospheric processes to refine deployment methodologies and assess their climate impact. Furthermore, stronger injection rates of aerosols should be discussed, as we simulated the current technological state of the art, but not the theoretically suggested spraying rate, which is a factor of 10 larger than the rate tested here. Assuming a linear relationship between aerosol concentration and droplet activation,  $N_c$  could increase to  $10 \times (N_c^* - N_c^0) + N_c^0 = 190 \, \mathrm{cm}^{-3}$  for clouds affected by the surface sprayer. From (2) and (3), we get

$$\frac{\mathrm{d}\ln(A_c)}{\mathrm{d}\ln(N_c)} = \frac{1 - A_c}{3},$$

which yields  $A_c^* = 0.581$ , and an increase in rCRE of d(rCRE)=0.0044 compared to the 0.0021 shown in this study, assuming the same  $f_c^*$ . While this represents a doubling from the initial d(rCRE) for the surface sprayer, it is still small compared to the base rCRE=0.2711. This indicates that spraying a larger fraction of clouds should be considered as important as the increase of cloud reflectance of individual clouds for MCB of cumulus clouds.

Data availability. All data produced for this study is publicly available via the Zenodo repository https://doi.org/10.5281/zenodo.17011767.

Author contributions. JK and FH conceived the original conceptualization of the presented work. JK, DPH, and FH contributed to the discussions and interpretation of the results. JK wrote the original draft and DPH and FH contributed to the review and editing. FH provided the funding acquisition and project administration.

220

215 Competing interests. The authors declare that none of the authors have any competing interests.

Acknowledgements. J. Kainz and F. Hoffmann were supported by the Emmy Noether program of the German Research Foundation (DFG) under Grant HO 6588/1-1 and HO 6588/3-1. D. P. Harrison was supported by the Reef Restoration and Adaptation Program. The Reef Restoration and Adaptation Program is funded by the partnership between the Australian Governments Reef Trust and the Great Barrier Reef Foundation. The authors acknowledge the Gauss Centre for Supercomputing e.V. for providing computing time on the GCS Supercomputer SuperMUC-NG at the Leibniz Supercomputing Centre (LRZ). The soundings were supplied by University of Wyoming (http://www.weather.uwyo.edu/upperair/sounding.shtml)

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
