# Peer review of "Marine Cloud Brightening of Cumulus Clouds: From the Sprayer to the Cloud"

_EGUsphere, 2025_

## Referee Comment (RC2)

The authors are performing an analysis of Marine Cloud Brightening (MCB) of trade-wind cumulus clouds from the BOMEX campaign. The authors use a Lagrangian Cloud Microphysics model, and three sets of simulations with aerosols sprayed at three different heights. The authors analyzed the horizontal and vertical spread of the aerosols, and the responses of cloud albedo, cloud fraction, cloud water mass and number concentration, and aerosol activation rates. The authors concluded that the near surface position is the ideal sprayer height, resulting in the highest brightening due to increased spreading in the sub-cloud turbulent layer, high availability of water vapor, and high aerosol activation rate.

The manuscript is well written and presents novel work that contributes significantly to scientific progress. The manuscript is suitable for publication with minor revisions. The manuscript needs to address three aspects in detail.

1. The grid spacing of 40m x 40m x 20m is likely too coarse for the MCB analysis. Dhandapani et al., cited in the manuscript, looked at sensitivity to grid spacing for MCB of stratocumulus clouds and concluded a horizontal grid spacing of 20m or lower is needed. The authors should include an analysis or justification for the grid spacing used here and the expected differences in simulations using finer grid spacing.

2. "This indicates that the increase in the average LWPc in sprayed clouds is not due to the spraying creating deeper clouds but due to the sprayed aerosol interacting preferentially with deeper clouds." The sprayed aerosol could also be interacting preferentially with brighter clouds and clouds with higher liquid water. To that end, the authors should include simulations with passive tracers released, to further separate correlation and causation. Since the passive tracers do not affect sub-cloud boundary layer or cloud properties, three different passive tracers could be released at the three different altitudes in the same simulation for efficiency.

3. The authors should discuss the radiation scheme used in the simulations, and possible consequences of the increased droplet concentration such as increased entrainment in cloud tops, decreased rain rate, changes in cloud longevity, etc.

Specific comments:

1. Repetition 'of the of the' in fig. 2 caption
2. Line 39: 'In Figs. 2a, c, and f' should be changed to 'In Figs. 2a, c, and e'